

# A message recovery attack on multivariate polynomial trapdoor function

Rashid Ali[1], Muhammad Mubashar Hussain[2], Shamsa Kanwal[3], Fahima Hajjej[4] and Saba Inam[3]

[1] Department of Mathematics, Capital University of Science and Technology, Islamabad, Pakistan
[2] Department of Mathematics, University of Punjab, Jhelum, Pakistan
[3] Department of Mathematical Sciences, Fatima Jinnah Women University, Rawalpindi, Rawalpindi, Pakistan
[4] Department of Information Systems, College of Computer and Information Sciences, Princess Nourah bint Abdulrahman University, Riyadh, Saudi Arabia

## ABSTRACT

Cybersecurity guarantees the exchange of information through a public channel in a secure way. That is the data must be protected from unauthorized parties and transmitted to the intended parties with confidentiality and integrity. In this work, we mount an attack on a cryptosystem based on multivariate polynomial trapdoor function over the field of rational numbers Q. The developers claim that the security of their proposed scheme depends on the fact that a polynomial system consisting of 2n (where n is a natural number) equations and 3n unknowns constructed by using quasigroup string transformations, has infinitely many solutions and finding exact solution is not possible. We explain that the proposed trapdoor function is vulnerable to a Gröbner basis attack. Selected polynomials in the corresponding Gröbner basis can be used to recover the plaintext against a given ciphertext without the knowledge of the secret key.

# INTRODUCTION

The 21$^{st}$ century is the century of information and technology. Because of the advancements in the field of information technology, the secure communication has become the most challenging task. Public key cryptography plays a vital role in this regard. The security of most of the public key cryptosystems being used is based on the intractability of certain mathematical problems which are considered to be hard. For instance, the security of RSA (*Rivest, Shamir & Adleman, 1978*) relies on the difficulty of *integer factorization problem* (IFP) and *ElGamal (1985)* is based on the hardness *discrete logarithm problem* (DLP). But these problems can be solved on quantum computers using Shor's algorithm (*Shor, 1997*). It is believed that multivariate public key cryptography is a good alternative in post-quantum reign for better security and efficiency. The security of a multivariate public key cryptosystems (MPKCs) relies on the difficulty of solving a system of multivariate polynomial equations (*Ding & Yang, 2009*) or isomorphism problem (*Tang & Xu, 2012*). In this context, several MPKCs were designed *e.g.*, Matsumoto-Imai multivariate quadratic polynomial scheme (*Matsumoto & Imai, 1988*), the Hidden Field Equation method

Corresponding author
Shamsa Kanwal,
shamsa.kanwal@fjwu.edu.pk

(*Patarin, 1996*), the Oil-Vinegar scheme (*Patarin, 1997*), *etc.* However, almost all of these schemes have been broken through various attacks (*Courtois, 2001*; *Faugère & Joux, 2003*; *Patarin, 1995*). A survey article on these schemes was written by *Wolf & Preneel (2005)*. *Markovski, Mileva & Dimitrova (2014)* have introduced a new multivariate polynomial trapdoor over the field of rational numbers.

Algebraic attacks (*Faugère & Joux, 2003*; *Kreuzer & galore!, 2009*) can be roughly divided into two categories. Firstly, the attacks which concentrate on specific variety and break it because of particular properties *e.g.*, *Kipnis & Shamir (1998)* attack against Oil and Vinegar. The second category comprises of algorithms generally used to solve multivariate polynomial system of equations. Examples include the XL algorithm (*Courtois et al., 2000*), and the relinearization technique (*Kipnis & Shamir, 1999*). *Buchberger (1965)* laid down a solid foundation of modern computational algebra by introducing the idea of Gröbner bases to address the problem of solving an algebraic system of multivariate polynomial equations. The Gröbner basis method is a general and well established technique to solve polynomial system of equations (*Buchberger, 1976*). For some applications of Göbner bases we refer to *Buchberger & Winkler (1998)*, *Buchberger (0000)* and *Francis & Ambedkar (2018)*. and for the detailed theory on computation of Gröbner basis we refer to the comprehensive books (*Cox, Little & O'shea, 1998*; *Kreuzer & Robbiano, 2000*) on computational algebra. The Buchberger's algorithm turns out to be very useful to mount an algebraic attack on any multivariate cryptosystem.

In this article, the cryptanalysis of a multivariate polynomial trapdoor function (*Markovski, Mileva & Dimitrova, 2014*) over the field of rational numbers is presented. The authors claimed that the proposed scheme is based on $2n$ multivariate polynomial equations in $3n$ unknowns and hence has infinitely many solutions to defeat an algebraic attack. Our cryptanalysis shows that the proposed multivariate scheme is vulnerable to Gröbner basis attack on the associated system of multivariate polynomial equations.

The rest of the article is organised as: 'Introduction' gives the brief description of the proposed scheme along with the necessary notations and definitions; 'The Multivariate Cryptosystem SBIM(Q)' illustrates the scheme with the example given in *Markovski, Mileva & Dimitrova (2014)*; 'Cryptanalysis' presents the cryptanalysis of the proposed scheme.

## THE MULTIVARIATE CRYPTOSYSTEM SBIM($\mathbb{Q}$)

The trapdoor function under consideration uses the multivariate polynomials, usually quadratic, over $\mathbb{Q}$, the field of rational numbers. The public key of this trapdoor function mainly consists of $2\,n$ multivariate polynomials in $3\,n$ unknowns $r_1, \ldots, r_n, s_1, \ldots, s_{2n}$. The variables $r_i; \mathrm{i}=1, \ldots, n$ usually contain the information content, whereas the variables $s_i; \mathrm{i}=1, \ldots, 2n$ contain the redundant information. The redundant information is added for the security purpose. So, if we use a plaintext comprising of $n$ rational numbers for the encryption purpose, we will get a ciphertext consisting of $2\,n$ rational numbers. The quasigroup string transformations are used to construct the public key. These transformations are obtained from quasigroups represented in matrix form. The private key of this cryptosystem comprises of different $1 \times n$ and $n \times n$ matrices over the field of rational numbers, and one $2n \times 2n$ matrix.

Recall that a groupoid $(G, f)$ having unique left as well as right inverses for each element in $G$ with respect to the binary operation $f$ is called a quasigroup. The binary operation $f : G \to G$ is then called a quasigroup bipermutation. From the binary operation $f$ on the quasigroup $G$ we can derive two new quasigroup bipermutation $f^{(23)}$ and $f^{(13)}$ as follows:

$$f(x_1, x_2) = x_3 \quad \Leftrightarrow \quad f^{(23)}(x_1, x_3) = x_2 \quad \Leftrightarrow \quad f^{(13)}(x_3, x_2) = x_1.$$

The next theorem gives a way to construct quasigroup bipermutation from matrices over a field $\mathbb{F}$.

**Theorem 2.1.** *(Markovski, Mileva & Dimitrova, 2014) Consider two nonsingular square matrices $A$ and $B$ of order $m$ over a field $\mathbb{F}$. Let $C$ be a row vector $(1 \times m$ matrix) over the field $\mathbb{F}$. Then the following mapping is a quasigroup bipermutation on $F^m$.*

$$f(r_1, \ldots, r_m; s_1, \ldots, s_m) = (r_1, \ldots, r_m) \cdot A + (s_1, \ldots, s_m) \cdot B + C, \tag{1}$$

*where $r_i, s_i \in F$. The new quasigroup bipermutations $f^{(13)}$ and $f^{(23)}$ are defined in the following way as:*

$$f^{(13)}(r_1, \ldots, r_m; s_1, \ldots, s_m) = (r_1, \ldots, r_m)A^{-1} + (s_1, \ldots, s_m)(-BA^{-1}) - CA^{-1},$$
$$f^{(23)}(r_1, \ldots, r_m; s_1, \ldots, s_m) = (r_1, \ldots, r_m)(-AB^{-1}) + (s_1, \ldots, s_m)B^{-1} - CB^{-1}. \tag{2}$$

Note that, in the above representation, instead of elements $r_i, s_i \in \mathbb{F}$, we can use polynomials $X_i$ and $r_i$ over $\mathbb{F}$ as inputs for the mapping $f$, then the output $f(X_1, \ldots, X_n; r_1, \ldots, r_n)$ will also be a polynomial.

## Construction

In this section we describe the construction of the proposed trapdoor multivariate public key cryptosystem (*Markovski, Mileva & Dimitrova, 2014*). From now on the field $\mathbb{F}$ is $\mathbb{Q}$, the field of rational numbers. A positive integer $n$ is used as a parameter of the scheme. The main global parameter is a multivariate polynomial ring in $3n$ indeterminates over the field of rational numbers $\mathbb{Q}$. The construction is based on three algorithms. That is, a Key Generation algorithm, an encryption algorithm and the corresponding decryption algorithm as described in the next sections. The message space is the set of all $n$-tuples $(a_1, \ldots, a_n) \in \mathbb{Q}$.

## Key generation

The key generation process comprises of the following steps:

1. **Choosing Polynomials:** Let $r_1, \ldots, r_n, s_1, \ldots, s_{2n}$ denote the variables on $\mathbb{Q}$. Choose $n$ multivariate polynomials $P_1, \ldots, P_n$ over $\mathbb{Q}$ in $n$ variables $r_1, \ldots, r_n$ in a way that the system of equations

   $$P_1(r_1, \ldots, r_n) = b_1,$$
   $$P_2(r_1, \ldots, r_n) = b_2,$$
   $$\ldots \quad \ldots \quad \ldots \tag{3}$$
   $$P_n(r_1, \ldots, r_n) = b_n,$$

   has a unique solution $r_1 = a_1, \ldots, r_n = a_n; a_i \in \mathbb{R}$ for any $b_i \in \mathbb{Q}$. Here, $\mathbb{R}$ denotes the field of real numbers. Next, choose $n$ more multivariate polynomials $P_{n+1}, \ldots, P_{2n}$ over $\mathbb{Q}$ with variables $r_1, \ldots, r_n, s_1, \ldots, s_{2n}$ over $\mathbb{Q}$.

2. **Applying Transformation:** First choose a random permutation $\tau$ on the set of integers $\{1, 2, \ldots, 2n\}$ and then apply it on $P_i$ to obtain the new polynomials $X_i$ such that $X_i = P_{\tau(i)}$ for all $i \in \{1, 2, \ldots, 2n\}$. Use these polynomials to define the vectors $\mathbf{x} = (X_1, \ldots, X_n)$ and $\mathbf{y} = (X_{n+1}, \ldots, X_{2n})$. Now $t-$ and $t'-$transformations are applied to obtain new polynomials as follows:

   (a) Define $t-$transformation: Choose a random vector $\mathbf{l}_1 = (\ell_{11}, \ldots, \ell_{1n}) \in \mathbb{Q}^n$ known as *leader* and then define two quasigroup bipermutations $f_1$ and $f_2$ by randomly choosing non singular $n \times n$ matrices $M_i, N_i$ $(i = 1, 2)$ as follows:
   $$f_1(\mathbf{l}_1; \mathbf{x}) = \mathbf{l}_1 \cdot M_1 + \mathbf{x} \cdot N_1; \quad f_2(\mathbf{x}'; \mathbf{y}) = \mathbf{x}' \cdot M_2 + \mathbf{y} \cdot N_2, \tag{4}$$
   where $\mathbf{x}' = f_1(\mathbf{l}_1; \mathbf{x})$ and set $\mathbf{y}' = f_2(\mathbf{x}'; \mathbf{y})$.

   (b) Define $t'-$transformation: Use the vector $y'$ and another random leader $\mathbf{l}_2 \in \mathbb{Q}^n$ where $\mathbf{l}_2 = (\ell_{21}, \ldots, \ell_{2n})$ to define new quasigroup bipermutations $f_3$ and $f_4$ by randomly choosing non singular $n \times n$ matrices $M_i, N_i$ $(i = 3, 4)$ as follows:
   $$f_3(\mathbf{y}'; \mathbf{l}_2) = \mathbf{y}' \cdot M_3 + \mathbf{l}_2 \cdot N_3, \quad f_4(\mathbf{x}'; \mathbf{y}'') = \mathbf{x}' \cdot M_4 + \mathbf{y}'' \cdot N_4, \tag{5}$$
   where $\mathbf{y}'' = f_3(\mathbf{y}'; \mathbf{l}_2)$. Again set $\mathbf{x}'' = f_4(\mathbf{x}'; \mathbf{y}'')$.

   These two $t-$ and $t'-$ transformations are necessary. Continuing this way, we can define more pairs of $t-, t'-$ quasigroup bipermutations from $\mathbf{y}''$ and $\mathbf{x}''$ by choosing new leaders $\mathbf{l}_i \in \mathbb{Q}^n$ and $n \times n$ random matrices $N_i, M_i$ in the same way as in the above Eqs. (4) and (5).

3. **The Public Key:** Let the integer $s \geq 0$ be the number of additional transformations applied. Note that the last transformation was accomplished by randomly chosen leader $\mathbf{l}_{2+p}$ and quasigroup bipermutations $f_{3+s}$ and $f_{4+s}$ applied on some $n-$tuples of multivariate polynomials $\mathbf{v}$ and $\mathbf{w}$. When the last applied transformation was a $t-$transformation, we write $f_{3+s}(\mathbf{l}_{2+s}; \mathbf{v}) :- (A_1, \ldots, A_n)$ and $f_{4+s}((A_1, \ldots, A_n); \mathbf{w}) :- (A_{n+1}, \ldots, A_{2n})$. Whereas if the last applied transformation was a $t'-$transformation, we let $f_{3+s}(\mathbf{w}; \mathbf{l}_{2+s}) :- (A_1, \ldots, A_n)$ and $f_{4+s}(\mathbf{v}; (A_1, \ldots, A_n)) :- (A_{n+1}, \ldots, A_{2n})$. Finally, choose a random non singular matrix $R$ over $\mathbb{Q}$ of order $2n \times 2n$ and compute the **public key**, $(Z_1, \ldots, Z_{2n})$, a new set of $2n$ polynomials as,
   $(Z_1, \ldots, Z_{2n}) = (A_1, \ldots, A_{2n}) \cdot R.$
   Clearly, each polynomial $Z_i = Z_i(r_1, \ldots, r_n, s_1, \ldots, s_{2n})$ is a multivariate polynomial in the $3n$ variables.

4. **The Private Key:** The permutation $\tau$, all the leaders $\mathbf{l}_i$ and all the matrices $M_i, N_i, R$, which were used to generate the public key, constitute the private key. Here we remark that all the leaders and the matrices are not necessarily required to be different but there should be at least two different leaders and at least four different matrices for defining the bipermutations.

## Encryption

To encrypt a message $M = (a_1, \ldots, a_n)$ in $\mathbb{Q}^n$, first choose $2n$ random rational numbers $b_1, \ldots, b_{2n}$ and then evaluate all the public polynomials $Z_i$ by setting $r_j = a_j$; j=1, ..., $n$ and $s_k = b_k$; k=1, ..., $2n$ to compute the ciphertext $\mathbf{c} = (c_1, \ldots, c_{2n})$. That is, the components of the ciphertext $\mathbf{c}$ are the rational numbers computed as follows:

$$c_1 \quad = Z_1(a_1, \ldots, a_n, b_1, \ldots, b_{2n}),$$

$$c_2 \;= Z_2(a_1,\ldots,a_n,b_1,\ldots,b_{2n}),$$

$$\ldots \quad \ldots \quad \ldots \quad \ldots \tag{6}$$

$$c_{2n} = Z_{2n}(a_1,\ldots,a_n,b_1,\ldots,b_{2n}).$$

## Decryption

To decrypt a ciphertext $\mathbf{c} = (c_1,\ldots,c_{2n})$, the receiver will first compute the inverse of the private matrix $R$ and compute the $2n$-tuple $(e_1,\ldots,e_{2n}) = (c_1,\ldots,c_{2n}) \cdot R^{-1}$ and split it into two halve to obtain $C_1 = (e_1,\ldots,e_n)$ and $C_2 = (e_{n+1},\ldots,e_{2n})$. Depending on how the polynomials $s_i$'s were obtained, the receiver has to then apply either a $u-$ or $u'-$transformation to undo the effect of $t-$ and $t'^{-}$transformations:

1. $u-transformation$: If the last transformation was a $t-$transformation defined by a leader $\mathbf{l}_{2+s}$ and bipermutations $f_{3+s}$ and $f_{4+s}$, then the receiver will apply a $u-$transformation defined by the parasstrophes $f_{3+s}^{(23)}$ and $f_{4+s}^{(23)}$ to obtain $M_1, M_2 \in \mathbb{Q}^n$ as follows:
   $$M_1 = f_{3+s}^{(23)}(\mathbf{l}_{2+s}; C_1), \quad M_2 = f_{4+s}^{(23)}(C_1; C_2).$$

2. $u'-transformation$: If the last transformation was a $t'-$transformation defined by a leader $\mathbf{l}_{2+s}$ and bipermutations $f_{3+s}$ and $f_{4+s}$, then the receiver will apply a $u-$ transformation defined by the parasstrophes $f_{3+s}^{(13)}$ and $f_{4+s}^{(13)}$ to obtain $M_1, M_2 \in \mathbb{Q}^n$ as follows:
   $$M_1 = f_{3+s}^{(13)}(C_2; \mathbf{l}_{2+s}), \quad M_2 = f_{4+s}^{(13)}(C_1; C_2).$$

Note that, we have to apply $u-$ or $u'-$transformations in the reverse order (from downward-up way). After each application of these transformations, we get $n-$tuples of rational numbers. In the end, instead of polynomial tuples $\mathbf{x}$ and $\mathbf{y}$ we get $n-$tuples of rational numbers $\mathbf{p} = (p_1,\ldots,p_n)$ and $\mathbf{q} = (p_{n+1},\ldots,p_{2n})$. Finally, the inverse permutation $\tau^{-1}$ is applied on $(p_1,p_2,\ldots,p_{2n})$ to get

$$(b_1 = p_{\tau^{-1}(1)},\ldots,b_{2n} = p_{\tau^{-1}(2n)}).$$

Use the values of $b_1,\ldots,b_n$ in the system Eq. (3) to get polynomial system of $n$ equations in $n$ unknowns. Solve the obtained system to get the required message $M = (a_1,\ldots,a_n) \in \mathbb{Q}^n$.

**Remark 2.2.** The trapdoor function described above takes plaintext in the form of $n-$tuple of rational numbers as input and returns the corresponding ciphertext in the form of $2n-$tuple of rational numbers as output. For the further details we refer to *Markovski, Mileva & Dimitrova (2014)*.

## CRYPTANALYSIS

The underlying hard problem in the above described multivariate trapdoor cryptosystem is that a polynomial system of equations consisting of $2n$ equations in $3n$ unknowns has infinite number of solutions. Therefore, finding the exact solution is not possible. For a given ciphertext $(c_1,\ldots,c_{2n})$ the attacker can make the following system using the public key polynomials $(s_1,\ldots,s_{2n})$.

$$Z_1(r_1,\ldots,r_n,s_1,\ldots,s_{2n}) \;= c_1,$$

$$Z_2(r_1,\ldots,r_n,s_1,\ldots,s_{2n}) \quad = c_2,$$
$$\ldots \quad \ldots \quad \ldots \tag{7}$$
$$Z_{2n}(r_1,\ldots,r_n,s_1,\ldots,s_{2n}) = c_{2n}.$$

The authors claim that, if the public key is produced by choosing suitable polynomials then the above system (Eq. (7)) has infinitely many solutions for the unknowns $r_1,\ldots,r_n$ and $s_1,\ldots,s_{2n}$. Therefore, an attacker cannot find the actual plaintext in this way. They proposed that using quadratic polynomials for $n = 4$, a much secure key can be generated. Here, we try different attacks to check its security. First of all, it is obvious that the private key consists of several matrices over the field of rational numbers and certain quasigroup bipermutations which shows that the key space is infinite. So the brute force attack is not possible even if the degree of the polynomials is known. Before we introduce the Gröbner bases attack method on this trapdoor function, note that, an attacker is not interested in all $3n$ unknowns. To recover the message $M = (a_1,\ldots,a_n)$ the attacker is only interested in the values of unknowns $r_i$ ($i = 1,\ldots,n$) containing the information. That is, to recover the message we do not have to solve the entire system of $2n$ equations in $3n$ unknowns.

Gröbner bases method is based on the Buchberger's algorithm (*Buchberger, 1965*) which is used to calculate Gröbner bases $G$ for the ideal $I$ generated by the polynomials in the system to be solved. Let $\mathbb{F}$ be a field and $I \subset \mathbb{F}[r_1,\ldots,r_n]$ be an ideal generated by the polynomials $f_1,\ldots,f_v \in \mathbb{F}[r_1,\ldots,r_n]$. Then a set $G = \{g_1,\ldots,g_k\} \subset I$ will be a Gröbner bases for $I$ with respect to some monomial ordering $\prec$ if the ideal generated by the leading terms of $G$ is the same as the ideal generated by the leading terms of $I$. For a given monomial ordering, every ideal has a Gröbner bases (for details, see *Cox, Little & O'shea, 1998*; *Kreuzer & Robbiano, 2000*).

## THE ATTACK MODEL

As stated earlier, the attacker is not interested in the infinitely many solutions of a system of $2n$ polynomial equations in $3n$ unknowns. One can exploit the structure of the multivariate cryptosystem presented in Construction 2.1 to mount a Gröbner basis attack by extracting a system of $n$ polynomials depending only in in $n$ unknowns $r_1,\ldots,r_n$ from the resulting Gröbner basis.

To mount the proposed attack, set the working ring $\mathbb{Q}[r_1,\ldots,r_n,s_1,\ldots,s_{2n}]$ of $3n$ indeterminates defined over the field of rational numbers $\mathbb{Q}$. After getting the public key polynomials $Z_1, Z_2,\ldots,Z_{2n}$ and the ciphertext $C = (c_1,c_2,\ldots,c_{2n}) \in \mathbb{Q}^{2n}$, perform the steps in the following attack for the cryptanalysis of the cryptosystem described in Section 2.

**Attack 3.1.** (Message Recovery Attack)

**Input:** Public key polynomials $Z_1,\ldots,Z_{2n} \in \mathbb{Q}[r_1,\ldots,r_n,s_1,\ldots,s_{2n}]$ and
Ciphertext $C = (c_1,c_2,\ldots,c_{2n}) \in \mathbb{Q}^{2n}$.

**Output:** A system of $n$ polynomial equations in $n$ unknowns.

**Step 1.** Create an ideal $I \subset \mathbb{Q}[r_1,\ldots,r_n,s_1,\ldots,s_{2n}]$ as
$$I = \langle Z_1 - c_1, Z_2 - c_2,\ldots,Z_{2n} - c_{2n}\rangle.$$

**Step 2.** Compute the reduced Gröbner basis $G = \{g_1,\ldots,g_t\} \subset \mathbb{Q}[r_1,\ldots,r_n,s_1,\ldots,s_{2n}]$ of $I$.

**Step 3.** Identify the polynomials $G_1, \ldots, G_n \in G$ depending only on the variables $r_1, \ldots, r_n$. That is, $G_i = g_j$ for some $g_j \in G$ such that $g_j \in \mathbb{Q}[r_1, \ldots, r_n]$.

**Step 4.** Solve the polynomial system of $n$ equations $\{G_1 = 0, \ldots, G_n = 0\}$ for the values of $r_1, \ldots, r_n$ to recover the message $M$.

Note that, the success of Attack heavily depends on the successful execution of Step 2 of the attack. We have already noticed that the construction of public polynomials is based on the constant multiples of the $n$ secret polynomials $P_1, \ldots, P_n$ depending only on the variables $r_1, \ldots, r_n$. Therefore, the resulting Gröbner basis will always contain polynomials depending only on these variables.

We now illustrate Attack 3.1 by mounting it first on the instance of the cryptosystem for $n = 2$ as given in [18, Section 4] and then for the case of $n = 4$.

**Example 3.2** Using our notations and symbols given in Section Section 2, we use the information presented in encryption example of *Markovski, Mileva & Dimitrova (2014)* to mount the attack as follows. Here we have $n = 2$ and the resulting public key consists of the following 4 polynomials $Z_1, \ldots, Z_4$ in $3n = 6$ unknowns $(r_1, r_2, s_1, s_2, s_3, s_4)$.

$$Z_1 = -8 + 7r_1 + 13r_2 - 9s_1 - 9s_4 + 11r_1^3 + 9r_2^3 + 9s_1^3 + 27r_1s_4 + 18r_2s_2$$

$$Z_2 = 21 - 5r_1 + 4r_2 + 9s_1 - 3s_3 + 3s_4 - 9r_1^3 - 6r_2^3 - 6s_1^3 - 3s_2^4 - 18r_1s_4 - 3r_2s_1 - 12r_2s_2$$

$$Z_3 = -10 - 5r_1 + r_2 + 3s_1 + 3s_4 + r_1^3 - 3r_2^3 - 3s_1^3 - 9r_1s_4 - 6r_2s_2$$

$$Z_4 = 13 - 9r_1 - 18r_2 + 12s_1 + 12s_4 - 16r_1^3 - 12r_2^3 - 12s_1^3 - 36r_1s_4 - 24r_2s_2.$$

This public key has been produced by the key generation process given in Section (2.2) with the following polynomials:

$$P_1 = r_1 - 2r_2,$$
$$P_2 = r_1^3 - 2,$$
$$P_3 = r_1^3 + r_2^2 + s_1^3 + 3r_1s_4 + 2r_2s_2 + r_1 + r_2 - s_1 - s_4,$$
$$P_4 = -s_2^4 - r_2s_1 + 2r_1 + s_1 - s_3 - s_4.$$

For the construction, the random permutation is taken as $\tau = (3, 2, 1, 4)$. The secret matrices involved in transformation Eqs. (4) and (5) are chosen as:

$$M_1 = \begin{pmatrix} 1 & -1 \\ 2 & -1 \end{pmatrix}, M_2 = \begin{pmatrix} -1 & 0 \\ 1 & 1 \end{pmatrix}, M_3 = \begin{pmatrix} -1 & 0 \\ 0 & -1 \end{pmatrix}, M_4 = \begin{pmatrix} 1 & -2 \\ 1 & 1 \end{pmatrix}$$

$$N_1 = \begin{pmatrix} 0 & 3 \\ 1 & 0 \end{pmatrix}, N_2 = \begin{pmatrix} 2 & 1 \\ -1 & -1 \end{pmatrix}, N_3 = \begin{pmatrix} 3 & 5 \\ 1 & 2 \end{pmatrix}, N_4 = \begin{pmatrix} 1 & 0 \\ 0 & 1 \end{pmatrix}$$

The leaders involved are $l_1 = (-1, 1)$ and $l_2 = (2, -1)$. Finally, a the invertible matrix $R$ of order $2n = 4$ is chosen as $R = \begin{pmatrix} 2 & -1 & 0 & -3 \\ 1 & 2 & -1 & -1 \\ 0 & 3 & 2 & 0 \\ -3 & -1 & -1 & 4 \end{pmatrix}$.

The message $M = (1, 1) \in \mathbb{Q}^2$ is encrypted by evaluating the public polynomials at $r_1 = 1$, $r_2 = 1$ and 4 randomly chosen rational numbers $s_1 = s_2 = s_3 = 0$, $s_4 = 1$. That is, the

resulting ciphertext $(c_1, c_2, c_3, c_4)$ is computed as:

$$c_1 = Z_1(1,1,0,0,0,1) = 50$$
$$c_2 = Z_2(1,1,0,0,0,1) = -10$$
$$c_3 = Z_3(1,1,0,0,0,1) = -22$$
$$c_4 = Z_4(1,1,0,0,0,1) = -66$$

With this ciphertext $C = (50, -10, -22, -66)$, we want to recover the corresponding plaintext $M = (1,1)$ without usin the secret key. For this purpose, we construct the following system of equations by using the public key polynomials $Z_1, Z_2, Z_3$ and $Z_4$ and the ciphertext.

$$Z_1 - 50 = 0,$$
$$Z_2 + 10 = 0,$$
$$Z_3 + 22 = 0,$$
$$Z_4 + 66 = 0.$$

To mount the Gröbner basis attack, let $I = \langle Z_1 - c_1, Z_2 + c_2, Z_3 - c_3, Z_4 - c_4 \rangle$ be the ideal generated by the above system of multivariate polynomial system of equations.

We use the computer algebra system ApCoCoA (*ApCoCoA Team, 2023*) and the code given in Appendix A for calculating the reduced Gröbner bases $G$ for the ideal $I$. The set $G$ is found to contain the following four polynomials:

$$F_1 = r_1 - 2r_2 + 1,$$
$$F_2 = s_1^3 + \frac{3}{2}r_2^2 + 2r_2 s_2 + 6r_2 s_4 + \frac{9}{4}r_2 - s_1 - 4s_4 - \frac{23}{4},$$
$$F_3 = s_2^4 + r_2 s_1 - 4r_2 - s_1 + s_3 + s_4 + 3,$$
$$F_4 = r_2^3 - \frac{3}{2}r_2^2 + \frac{3}{4}r_2 - \frac{1}{4}.$$

Recall that the variables $r_i$'s contain the information about the original message while $s_i$ are the redundant variables. In the above computed Gröbner basis, we are only interested in polynomials $F_1$ and $F_4$ that are expressed in two required unknowns $r_1$ and $r_2$. Solving $F_1 = 0$ and $F_4 = 0$ simultaneously, the only real solution of $F_4 = 0$ is $r_2 = 1$, and $F_1 = 0$ then gives $r_1 = 1$. This shows that the plaintext $M = (r_1, r_2) = (1,1)$ has been successfully recovered without using the private key.

**Remark 3.3.** All computations are performed on the platform of Computer Algebra System ApCoCoA (*ApCoCoA Team, 2023*). For this purpose the Key Generation Algorithm 2.2 and the Encryption Algorithm 2.3 are implemented in the setting of ApCoCoA as given in Appendix A. The validity of the findings follows from the fact that our code generated the same public polynomials $P_1, P_2, P_3, P_4$ and the ciphertext $C = (c_1, c_2, c_3, c_4)$ as given in *Markovski, Mileva & Dimitrova (2014)*. Moreover, the computation of reduced Gröbner basis of the ideal $I$ has been performed by the built-in function `ReducedGBasis(I)` available in ApCoCoA (*ApCoCoA Team, 2023*).

**Example 3.4.** For the case of $n = 4$, the multivariate ring over $\mathbb{Q}$ in $3n = 12$ indeterminates is $\mathbb{Q}[r_1, \ldots, r_4, s_1, \ldots, s_8]$. As per requitremnt of the cryptosystem presented in *Markovski, Mileva & Dimitrova (2014)* the following secret polynomials $P_i \quad 1 \leq i \leq 8$ are chosen such that the polynomial system $\{P_1 = 0, P_2 = 0, P_3 = 0, P_4 = 0\}$ has unique solution $(a_1, a_2, a_3, a_4) \in \mathbb{Q}^4$.

$$P_1 = 5r_1 + r_2 + r_3 + 2r_4 + 1$$

$$P_2 = -r_1^2 - r_2 - r_4$$

$$P_3 = -r_1 + 3r_2 + 2$$

$$P_4 = -r_1 r_3 + 2r_1 + r_2 + 1$$

$$P_5 = r_1^2 + 5r_2^2 + s_1^2 + 2r_2 s_2 + 3r_1 s_4 + r_1 s_5 + r_2 s_5 + r_1 - r_2 - s_4 - s_5$$

$$P_6 = 2r_1 r_3 - 2r_2 s_1 - r_3 s_1 - s_2^2 + s_2 s_3 - s_4 s_5 + s_5 s_6 + r_4 s_7 - s_8^2 + 3s_5 - s_7 + s_8$$

$$P_7 = -2r_2 s_2 + r_3 s_3 + s_3^2 - 5s_4^2 + s_5 s_6 - 2s_5 s_7 + r_1 + 4r_4 + 2s_1 - s_3 - s_5 + s_7 + s_8 + 2$$

$$P_8 = r_2^2 + 5r_3^2 + 5r_1 s_1 - 3s_2^2 - 3r_2 s_4 + 2r_4 s_4 + s_4^2 + r_2 s_6 + r_3 s_7 + s_7^2 + s_8^2 + r_2 - r_3 + 2r_4 - s_5 - s_7$$

The random permutation is taken as $\tau = (3, 5, 1, 6, 8, 2, 4, 7)$ and for the transformations Eqs. (4) and (5), the secret matrices are taken as:

$$M_1 = \begin{pmatrix} 1 & -1 & 2 & 1 \\ 2 & -1 & 3 & 1 \\ 1 & 0 & 0 & -1 \\ 2 & 1 & 0 & 0 \end{pmatrix}, M_2 = \begin{pmatrix} -1 & 0 & 0 & 0 \\ 1 & 1 & 1 & 2 \\ 0 & 1 & -1 & 0 \\ 0 & 0 & 1 & 2 \end{pmatrix}, M_3 = \begin{pmatrix} -1 & 0 & 3 & 4 \\ 0 & -1 & 0 & -3 \\ 2 & 1 & 3 & 1 \\ 1 & 0 & 0 & 1 \end{pmatrix}, M_4 = \begin{pmatrix} 1 & 0 & 0 & 0 \\ 0 & 1 & 0 & 1 \\ 0 & 0 & -1 & 0 \\ 0 & 0 & 0 & 1 \end{pmatrix}.$$

$$N_1 = \begin{pmatrix} 1 & 3 & 0 & 3 \\ 1 & 0 & 1 & 0 \\ 2 & 1 & -1 & 2 \\ 3 & 0 & 0 & 1 \end{pmatrix}, N_2 = \begin{pmatrix} 2 & 1 & 1 & -1 \\ -1 & -1 & 3 & 4 \\ 1 & 0 & -1 & 0 \\ 2 & 3 & 5 & 1 \end{pmatrix}, N_3 = \begin{pmatrix} 3 & 5 & 1 & -1 \\ 1 & 2 & 1 & 0 \\ 0 & 0 & 2 & 0 \\ 2 & 0 & 3 & 0 \end{pmatrix}, N_4 = \begin{pmatrix} 1 & 0 & 0 & 0 \\ 0 & 1 & 0 & 0 \\ 0 & 0 & 1 & 0 \\ 0 & 0 & 0 & 1 \end{pmatrix}.$$

The leaders $\mathbf{l}_1, \mathbf{l}_2$ and random secret matrix $R$ are

$$\mathbf{l}_1 = (-1, 1, -1, 1), \mathbf{l}_2 = (2, -1, -2, 2), \text{ and } R = \begin{pmatrix} 2 & -1 & 0 & -3 & 1 & 0 & -1 & 3 \\ 1 & 2 & -1 & -1 & 1 & 5 & 0 & 0 \\ 0 & 3 & 2 & 0 & 0 & 0 & 3 & 2 \\ -3 & -1 & -1 & 4 & 4 & 2 & 1 & -1 \\ 1 & -1 & 2 & -2 & 3 & 1 & 4 & 5 \\ 1 & 4 & 5 & -1 & 0 & 0 & 0 & -2 \\ 0 & 1 & 0 & -3 & -3 & -1 & 2 & 0 \\ 1 & 1 & -1 & -1 & 1 & -1 & 2 & -1 \end{pmatrix}.$$

The resulting public polynomials are:

$$Z_1 = -18r_1^2 + 42r_2^2 + 88r_1 r_3 - 65r_3^2 - 65r_1 s_1 - 88r_2 s_1 - 44r_3 s_1 + 11s_1^2 - 20r_2 s_2$$
$$- 5s_2^2 + 21r_3 s_3 + 44s_2 s_3 + 21s_3^2 + 33r_1 s_4 + 39r_2 s_4 - 26r_4 s_4 - 118s_4^2 + 11r_1 s_5 + 11r_2 s_5 - 44s_4 s_5$$
$$- 13r_2 s_6 + 65s_5 s_6 - 13r_3 s_7 + 44r_4 s_7 - 42s_5 s_7 - 13s_7^2 - 57s_8^2 + 158r_1 + 42r_2 + 67r_3 + 103r_4 + 42s_1$$
$$- 21s_3 - 11s_4 + 113s_5 - 10s_7 + 65s_8 + 187,$$

$$Z_2 = -64r_1^2 - 152r_2^2 - 72r_1 r_3 + 190r_3^2 + 190r_1 s_1 + 72r_2 s_1 + 36r_3 s_1 - 38s_1^2 - 232r_2 s_2 - 78s_2^2$$

$$+78r_3s_3 - 36s_2s_3 + 78s_3^2 - 114r_1s_4 - 114r_2s_4 + 76r_4s_4 - 352s_4^2 - 38r_1s_5 - 38r_2s_5 + 36s_4s_5 + 38r_2s_6$$
$$+42s_5s_6 + 38r_3s_7 - 36r_4s_7 - 156s_5s_7 + 38s_7^2 + 74s_8^2 + 121r_1 + 287r_2 - 8r_3 + 422r_4 + 156s_1$$
$$-78s_3 + 38s_4 - 186s_5 + 76s_7 + 42s_8 + 341,$$

$$Z_3 = -46r_1^2 - 34r_2^2 + 46r_1r_3 + 30r_3^2 + 30r_1s_1 - 46r_2s_1 - 23r_3s_1 - 8s_1^2 - 132r_2s_2 - 41s_2^2 + 58r_3s_3$$
$$+23s_2s_3 + 58s_3^2 - 24r_1s_4 - 18r_2s_4 + 12r_4s_4 - 284s_4^2 - 8r_1s_5 - 8r_2s_5 - 23s_4s_5 + 6r_2s_6 + 81s_5s_6$$
$$+6r_3s_7 + 23r_4s_7 - 116s_5s_7 + 6s_7^2 - 17s_8^2 + 207r_1 + 193r_2 + 60r_3 + 306r_4 + 116s_1 - 58s_3 + 8s_4$$
$$+13s_5 + 29s_7 + 81s_8 + 347,$$

$$Z_4 = 63r_1^2 - 7r_2^2 - 102r_1r_3 - 35r_3^2 - 35r_1s_1 + 102r_2s_1 + 51r_3s_1 + 194r_2s_2 + 72s_2^2 - 97r_3s_3 - 51s_2s_3$$
$$-97s_3^2 + 21r_2s_4 - 14r_4s_4 + 478s_4^2 + 51s_4s_5 - 7r_2s_6 - 148s_5s_6 - 7r_3s_7 - 51r_4s_7 + 194s_5s_7 - 7s_7^2$$
$$+44s_8^2 - 361r_1 + 362r_2 - 106r_3 - 513r_4 - 194s_1 + 97s_3 - 49s_5 - 39s_7 - 148s_8 - 616,$$

$$Z_5 = -87r_1^2 - 141r_2^2 + 18r_1r_3 - 30r_3^2 - 30r_1s_1 - 18r_2s_1 - 9r_3s_1 - 27s_1^2 - 54r_2s_2 + 9s_2^2 + 9s_2s_3$$
$$-81r_1s_4 + 18r_2s_4 - 12r_4s_4 - 6s_4^2 - 27r_1s_5 - 27r_2s_5 - 9s_4s_5 - 6r_2s_6 + 9s_5s_6 - 6r_3s_7 + 9r_4s_7$$
$$-6s_7^2 - 15s_8^2 + 255r_1 + 441r_2 + 86r_3 + 92r_4 + 27s_4 + 60s_5 - 3s_7 + 9s_8 + 386,$$

$$Z_6 = -44r_1^2 - 70r_2^2 + 22r_1r_3 - 25r_3^2 - 25r_1s_1 - 22r_2s_1 - 11r_3s_1 - 13s_1^2 - 32r_2s_2 + 4s_2^2 + 3r_3s_3$$
$$+11s_2s_3 + 3s_3^2 - 39r_1s_4 + 15r_2s_4 - 10r_4s_4 - 20s_4^2 - 13r_1s_5 - 13r_2s_5 - 11s_4s_5 - 5r_2s_6 + 14s_5s_6$$
$$-5r_3s_7 + 11r_4s_7 - 6s_5s_7 - 5s_7^2 - 16s_8^2 + 138r_1 + 186r_2 + 53r_3 + 57r_4 + 6s_1 - 3s_3 + 13s_4 + 48s_5$$
$$-3s_7 + 14s_8 + 208,$$

$$Z_7 = -142r_1^2 - 248r_2^2 - 76r_1r_3 + 285r_3^2 + 285r_1s_1 + 76r_2s_1 + 38r_3s_1 - 61s_1^2 - 416r_2s_2 - 133s_2^2$$
$$+147r_3s_3 - 38s_2s_3 + 147s_3^2 - 183r_1s_4 - 171r_2s_4 + 114r_4s_4 - 678s_4^2 - 61r_1s_5 - 61r_2s_5 + 38s_4s_5$$
$$+57r_2s_6 + 109s_5s_6 + 57r_3s_7 - 38r_4s_7 - 294s_5s_7 + 57s_7^2 + 95s_8^2 + 404r_1 + 875r_2 + 55r_3 + 845r_4$$
$$+294s_1 - 147s_3 + 61s_4 - 257s_5 + 128s_7 + 109s_8 + 929,$$

$$Z_8 = -79r_1^2 + 5r_2^2 + 136r_1r_3 - 136r_2s_1 - 68r_3s_1 + s_1^2 - 198r_2s_2 - 68s_2^2 + 100r_3s_3 + 68s_2s_3$$
$$+100s_3^2 + 3r_1s_4 - 500s_4^2 + r_1s_5 + r_2s_5 - 68s_4s_5 + 168s_5s_6 + 68r_4s_7 - 200s_5s_7 - 68s_8^2 + 479r_1$$
$$+557r_2 + 151r_3 + 566r_4 + 200s_1 - 100s_3 - s_4 + 103s_5 + 32s_7 + 168s_8 + 793.$$

The message $M = (15, 10, 2, 3) \in \mathbb{Q}^4$ is encrypted by evaluating the public key polynomials at $r_1 = 15, r_2 = 10, r_3 = 2, r_4 = 3$ and 8 randomly chosen rational numbers $s_1 = 0, s_2 = 1, s_3 = 2, s_4 = 0, s_5 = -10, s_6 = 2, s_7 = 5, s_8 = 1$. The encryption scheme ?? resulted in the ciphertext $(c_1, c_2, \ldots, c_8)$ as given below:

$$c_1 = Z_1(15, 10, 2, 3, 0, 1, 2, 0, -10, 2, 5, 1) = 3258$$

$$c_2 = Z_2(15, 10, 2, 3, 0, 1, 2, 0, -10, 2, 5, 1) = -6360$$

$$c_3 = Z_3(15, 10, 2, 3, 0, 1, 2, 0, -10, 2, 5, 1) = 585$$

Ali et al. (2023), *PeerJ Comput. Sci.*, DOI 10.7717/peerj-cs.1521

$$c_4 = Z_4(15, 10, 2, 3, 0, 1, 2, 0, -10, 2, 5, 1) = -8226$$
$$c_5 = Z_4(15, 10, 2, 3, 0, 1, 2, 0, -10, 2, 5, 1) = -19001$$
$$c_6 = Z_4(15, 10, 2, 3, 0, 1, 2, 0, -10, 2, 5, 1) = -9398$$
$$c_7 = Z_4(15, 10, 2, 3, 0, 1, 2, 0, -10, 2, 5, 1) = -9244$$
$$c_8 = Z_4(15, 10, 2, 3, 0, 1, 2, 0, -10, 2, 5, 1) = 8465.$$

With this ciphertext $C = (3258, -6360, 585, -8226, -19001, -9398, -9244, 8465)$, we want to recover the corresponding plaintext $M = (15, 10, 2, 3)$ without using the secret key. For this purpose, we construct the following system of equations by using the public key polynomials $Z_1, Z_2, \ldots, Z_8$ and the ciphertext $C$.

$$Z_1 - 3258 = 0, \quad Z_2 + 6360 = 0, \quad Z_3 - 585 = 0, \quad Z_4 + 8226 = 0,$$
$$Z_5 + 19001 = 0, \quad Z_6 + 9398 = 0, \quad Z_7 + 9244 = 0, \quad Z_8 - 8465 = 0.$$

To mount Attack 3.1, let $I = \langle Z_1 - c_1, Z_2 + c_2, \ldots, Z_8 - c_8 \rangle$ be the ideal generated by the above system of multivariate polynomial equations. Using the computer algebra system (*ApCoCoA Team, 2023*), the reduced Gröbner basis $G$ of the ideal $I$ is computed. The computed Gröbner basis $G$ contains a total of 34 multivariate polynomials and of these polynomials, the following five polynomials are depending only on the variables of interest, that is, $r_1, \ldots, r_4$.

$$G_1 = r_3^2 + \frac{8971}{24}r_3 - \frac{15}{4}r_4 - \frac{2221}{3},$$
$$G_2 = r_1 + \frac{3}{16}r_3 + \frac{3}{8}r_4 - \frac{33}{2},$$
$$G_3 = r_2 + \frac{1}{16}r_3 + \frac{1}{8}r_4 - \frac{21}{2},$$
$$G_4 = r_3 r_4 - \frac{3713}{16}r_3 - \frac{11}{24}r_4 + \frac{919}{2},$$
$$G_5 = r_4^2 + \frac{27121}{288}r_3 - \frac{11575}{144}r_4 + \frac{1577}{36}.$$

To recover the message $M \in \mathbb{Q}^4$, solve the system

$$\{G_1 = 0, G_2 = 0, G_3 = 0, G_4 = 0, G_5 = 0\}. \tag{8}$$

Label the variables $r_1, r_2, r_3$, and $r_4$ by $x, y, z$, and $w$ respectively and then use online polynomial system solver by Wolfram (available at https://www.wolframalpha.com/calculators/equation-solver-calculator). The only rational solution of the polynomial system Eq. (8) is given below:

$$r_1 = x = 15, r_2 = y = 10, r_3 = z = 2, \text{ and } r_4 = w = 3.$$

Hence, the message $M = (15, 10, 2, 3)$ is successfully recovered by mounting the attack.

**Remark 3.5.** We have observed that the proposed cryptosystem is vulnerable to the Gröbner bases attack. The bipermutations used to produce the public key are linear in which the polynomials are not multiplied with each other. This can be the weakest part of

its construction. Because using linear bipermutations the Gröbner bases will contain the polynomials separately in the variables as were the starting polynomials. Among these, the polynomials in informative variables can be solved to get plaintext. The main cost in this attack is the Gröbner bases computation.

## COMPLEXITY ANALYSIS

As stated earlier that the success of Attack 3.1, depends on the computation of Gröbner basis of the ideal of interest. It is also known that the upper bound for the complexity of finding the solutions of a multivariate polynomial system with the help of the computation of Gröbner basis is a function of the degree of regularity $d_{\text{reg}}$, the maximum degree observed during the process of computation. In the worst case scenario, this complexity is known to be doubly exponential in number of variables $n$, for details see (*Bardet, Faugère & Salvy, 2015*) and the references therein. This means that, in general or random setting, finding Gröbner basis is not an easy job. However, in the present scenario, to leave a trapdoor for the multivariate polynomial cryptosystem under consideration, the polynomials $\{P_1, \ldots, P_n\}$ are special in the sense that the system of equations Eq. (3) should has a unique solution $(r_1, \ldots, r_n) = (a_1, \ldots, a_n) \in \mathbb{Q}^n$ for all choices of the constants $b_i$'s.

Moreover, for the secure instances of the cryptosystem, the authors suggested that the value of $n = 4$ is safe to choose . Therefore, in any such instance, there will be $2n = 8$ polynomials in $3n = 12$ variables $r_1, \ldots, r_4, z_1, \ldots, z_8$. Out of these 8 polynomials, four polynomials $P_1, \ldots, P_4$ are depending only on 4 variables $r_1, \ldots, r_4$. For the required trapdoor in the construction presented in Construction 2.1, one has to start by choosing these four polynomials in such a way that the system

$$P_1(r_1, \ldots, r_4) = b_1,$$
$$P_2(r_1, \ldots, r_4) = b_2,$$
$$P_3(r_1, \ldots, r_4) = b_3, \tag{9}$$
$$P_4(r_1, \ldots, r_4) = b_4,$$

has a unique solution $(r_1, r_2, r_3, r_4) = (a_1, a_2, a_3, a_4) \in \mathbb{Q}^4$ for all choices of the constants $b_1, b_2, b_3$ and $b_4$. Later on, 4 more polynomials are constructed by involving all the 12 variables, making a system of 8 equations in 12 unknowns. The public key polynomials $\{Z_1, \ldots, Z_{3n}\}$ are then obtained by some random linear combinations of the polynomials $\{P_1, \ldots, P_{2n}\}$ by using bipermutations Eqs. (4) and (5). In the entire construction, only $n$ variables $r_1, \ldots, r_n$ are basic (or informative) and rest of the $2n$ variables $s_1, \ldots, s_{2n}$ are redundant.

The requirement of the unique solution of the system Eq. (9) makes the system Eq. (7) of $2n$ polynomials quite special rather than a general and hence the worst case scenario of the complexity of Gröbner basis computation is not applicable here. Moreover, we are not interested in the infinitely many solutions of the system Eq. (7) containing the values of the redundant unknowns $s_1, \ldots, s_{2n}$ but only the unknowns $r_1, \ldots, r_n$ are required to recover the message $M$. It, therefore, follows that there is no need to compute

the complete Gröbner basis of the ideal $I = \langle Z_1 - c_1, Z_2 + c_2, \ldots, Z_8 - c_8 \rangle$. One can terminate the Gröbner basis computation process when sufficient number of polynomials depending only on the basic variables are obtained. Again, the worst-case estimate of the complexity is not applicable.

This can also be achieved with the help of the well known application of the Gröbner basis, namely, the *elimination theory*. That is, just calculate the elimination ideal $I \cap \mathbb{Q}[r_1, \ldots, r_4]$ and then solve the system to recover the message.

Several instances of the multivariate cryptosystem as illustrated in Example 3.4 are computed for $n = 4$ and the message was successfully recovered by mounting Attack 3.1 and the Encryption Code (Appendix A) on the Dell laptop Latitude 3520 (11th Gen Intel(R) Core(TM) i5-1135G7 2.40 GHz, 8.0 GB Ram). For the computations involved in Example 3.4, the CPU time was recorded by ApCoCoA (*ApCoCoA Team, 2023*) as 7.2 sec. for the complete Gröbner basis computation. On the other hand, the total CPU time recorded as 285 millisecond by ApCoCoA in the computation of elimination ideal $J = I \cap \mathbb{Q}[r_1, r_2, r_3, r_4]$ and then computation of Gröbner basis of $J$. In many other instances with the parameter $n = 4$, the recorded time for the reduced Gröbner basis computation was within 2 sec. Therefore, the multivariate cryptosystem presented in Section Construction 2.1 is not secure against Gröbner basis Attack.

## CONCLUSION AND FUTURE WORK

In this article, we studied the security of the multivariate polynomial trapdoor public key cryptosystem proposed by *Markovski, Mileva & Dimitrova (2014)*. We found that although the public key consists of less polynomials than the number of variables which will result in infinite many solutions of the polynomial system, even then the cryptosystem does not seem to be secure. One can mount a Gröbner bases attack against the recommended parameter $n = 2$ and nonlinear multivariate polynomial system (Eq. (3)) to recover the message without the knowledge of the secret key. The attack successfully recovers the original message that was encrypted by this cryptosystem in Section 4 of *Markovski, Mileva & Dimitrova (2014)*. Moreover, the successful cryptanalysis of several other instances of this cryptosystem reveals that this cryptosystem is vulnerable to Gröbner bases attack. Moreover, the starting step in the key generation algorithms is to choose suitable polynomials in a way that the system (Eq. (3)) should have a unique solution. Although a linear system to meet this requirement can be constructed trivially but the construction of a nonlinear system of polynomial equations for $n \geq 4$ is not an easy task. Therefore, a concrete way should be provided to formulate a system having unique real solution to generate a strong public key; that is, a public key to produce a ciphertext which is secure against Gröbner bases attack. Hence, we conclude that there are many security flaws in the proposed multivariate cryptosystem.

# APPENDIX A. KEY GENERATION AND ENCRYPTION CODE

The following code is written in CoCoL language for the computer algebra system ApCoCoA (*ApCoCoA Team, 2023*). In the code below, the parameters are as stated below:

| Parameter | Description with reference to Section Construction 2.1. |
|---|---|
| N | The value of $n$ taken as 2 or 4 |
| PolyYs | The secret polynomials $P_1, \ldots, P_{2n}$ |
| Perm | The random permutation $\tau$ |
| MatAs | The matrices $M_1, \ldots, M_4$ |
| MatBs | The matrices $N_1, \ldots, N_4$ |
| Leads | The leaders $\mathbf{l}_1$ and $\mathbf{l}_2$ |
| Rmat | The random matrix $R$ of order $2n \times 2n$ |
| PlainTxtZ | The $3n$-tuple containing the message and the random $2n$ values of $s_1, \ldots, s_{2n}$ |

```
Define SBIMPK(N, PolyYs, Perm, MatAs, MatBs, Leads, Rmat, PlainTxtZ)
  PolyXs := [];
   For I := 1 To 2*N Do
     Append(PolyXs, PolyYs[Perm[I]]);
   EndFor;
  Vx := [];Vy := [];
   For I := 1 To N Do
     Append(Vx, PolyXs[I]);
     Append(Vy, PolyXs[I+N]);
   EndFor;
  Vx := Mat([Vx]);Vy := Mat([Vy]);
  Fxy := [];
   For I := 1 To 4 Do
     Append(Fxy, Vx*MatAs[I]+Vy*MatBs[I]);
   EndFor;
  —1st e-transformation use L1, Vx, and Vy to get Xd, Yd
  F1L1Vx := Leads[1]*MatAs[1]+Vx*MatBs[1]; –called it x-dash to use in F2
  F2f1Vx := F1L1Vx*MatAs[2]+Vy*MatBs[2]; –Called it y-dash to use it in F3
  —2nd e'-transformation use L2, x-dash, and y-dash
  F3f2L2 := F2f1Vx*MatAs[3]+Leads[2]*MatBs[3]; –Called it y-ddash
  F4f1f3 := F1L1Vx*MatAs[4]+F3f2L2*MatBs[4]; –called it x-ddash
  Zmat := []; –row vectro Z
   For I := 1 To N Do
     Append(Zmat, F4f1f3[1][I]);
   EndFor;
   For I := 1 To N Do
     Append(Zmat, F3f2L2[1][I]);
```

```
    EndFor;
   Zmat := Mat([Zmat]);
   PU := List(Zmat*Rmat);
   Cipher := Eval(PU, PlainTxtZ);
   Return (PU[1],Cipher[1]);
  EndDefine;
```

### Funding

This study was funded by the Princess Nourah bint Abdulrahman University Researchers Supporting Project number (PNURSP2023R236), Princess Nourah bint Abdulrahman University, Riyadh, Saudi Arabia. The funders had no role in study design, data collection and analysis, decision to publish, or preparation of the manuscript.

### Grant Disclosures

The following grant information was disclosed by the authors:
The Princess Nourah bint Abdulrahman University Researchers: PNURSP2023R236.
Princess Nourah bint Abdulrahman University, Riyadh, Saudi Arabia.

### Competing Interests

The authors declare that there are no competing interests.

### Author Contributions

- Rashid Ali conceived and designed the experiments, performed the computation work, authored or reviewed drafts of the article, and approved the final draft.
- Muhammad Mubashar Hussain performed the experiments, analyzed the data, performed the computation work, prepared figures and/or tables, and approved the final draft.
- Shamsa Kanwal analyzed the data, authored or reviewed drafts of the article, and approved the final draft.
- Fahima Hajjej analyzed the data, prepared figures and/or tables, and approved the final draft.
- Saba Inam performed the computation work, authored or reviewed drafts of the article, and approved the final draft.

### Data Deposition

No raw data was generated in this study.

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
