# Peer review of "A message recovery attack on multivariate polynomial trapdoor function"

_PeerJ Computer Science, doi:10.7717/peerj-cs.1521_

## Round 0.1 · original submission · Major Revisions

Based on the reviewers' comments, the paper needs major revision to be considered for publication in PeerJ Computer Science journal. When you make the changes, please, resubmit the paper, which will be returned to the reviewers. Thank you for considering publishing your work in PeerJ Computer Science journal.

Reviewer 1 ·

Basic reporting

The attack method seems promising, however it is known that
finding rational solutions of rational coefficient polynomial
systems is rather different from solving in a finite field,
and the authors of the cited paper should already consider
using Groebner basis as a potential attack, which they didn't.
Anyway, the paper does not contain any complexity analysis, so not ready for publications

Experimental design

Not sufficient

Validity of the findings

Not substantial enough

Reviewer 2 ·

Basic reporting

The manuscript proposes a message recovery attack on SBIM(Q), a multivariate polynomial trapdoor over the field of rational numbers Q, which was proposed by Markovski et al in 2014. SBIM(Q) encrypts a plaintext of n rational numbers into a ciphertext of 2n rational numbers, and its security is based on the complexity of 2n polynomial equations with 3n unknowns.

The authors of this manuscript demonstrate a successful cryptoanalysis on the SBIM(Q) trapdoor using a Grobner basis attack. They show that plaintext message recovery from the ciphertext can be achieved without knowing the secret key. The authors use a toy example from the original paper to demonstrate the cryptanalysis and find that some polynomials of its Grobner basis only consist of information variables. Those information-only polynomials in the Grobner basis can be used to retrieve the secret. The authors point out that the cost of this vulnerability in SBIM(Q) is due to the linear bipermutations, which are exploited by the Grobner basis attack.

The paper is well-written with sufficient information and background and good logic description.

Here are some comments for the authors to consider:
1. The section title “The Proposed Multivaraite Cryptosystem” causes confusion. It may be misunderstood the cryptosystem proposed by the authors. It is in fact from a cited reference.
2. There is only one sub-section in Section 2, it may be better to reorganize Key Gen, Encrypt and Decrypt.

Experimental design

There is no experimental design in this paper.

Validity of the findings

Their cryptoanalysis is interesting and sounds correct on the SBIM(Q).

Annotated reviews are not available for download in order to protect the identity of reviewers who chose to remain anonymous.

Reviewer 3 ·

Basic reporting

A review of the paper “A Message Recovery Attack on Multivariate Polynomial Trapdoor Function”
by the authors Rashid Ali, Muhammad Mubashar Hussain, Shamsa Kanwal, Fahima Hajjej and Saba Inam

Markovski et al. in the paper “SBIM(Q) - a Multivariate Polynomial Trapdoor unction over the Field of Rational Numbers” proposed a multivariate polynomial trapdoor function over the field of rational numbers Q. Its public key comprises of 2n polynomials in 3n unknowns and is constructed by using quasigroup string transformations. So, a plaintext consisting of n rational numbers will be encrypted to a ciphertext consisting of 2n rational numbers. The authors claim that the security of their proposed scheme depends on the fact that a polynomial system consisting of 2n equations and 3n unknowns has infinitely many solutions, so finding exact solution is not possible.
The authors of the paper under reviewing made a cryptanalysis of this trapdoor function by mounting a Groebner basis attack. They have shown that the proposed trapdoor function is vulnerable to a Groebner bases attack. Namely, selected polynomials in the corresponding Groebner basis can be used to recover the plaintext against a given ciphertext without the knowledge of the secret key.
The paper is written in clear and readable way and only a few typos can be found. Here are some:
42 … polynomial equations Groebner basis method …
… polynomial equations. Groebner basis method …
50 … Our cryptnalysis shows …
… Our cryptanalysis shows …
119 … order 2n x 2n compute …
… order 2n x 2n and compute …
129 … and the evaluate …
… and then evaluate …
186 … (r1, r2, s1, s2, s3, s3). …
… (r1, r2, s1, s2, s3, s4). …

My opinion is that the paper can be published in your journal.

Experimental design

ok

Validity of the findings

ok

Additional comments

no

Annotated reviews are not available for download in order to protect the identity of reviewers who chose to remain anonymous.

---

## Round 0.2 · accepted · Accept

Thank you for considering the reviewers' comments and making the required changes to your article.

I am happy to inform you that your paper now reaches the required level for publication in PeerJ Computer Science. Thank you for considering our journal for publishing your research papers.

We hope you will continue to consider the journal for publication in your future research.

Sincerely,

M. Emilia Cambronero
Academic Editor of PeerJ Computer Science

Reviewer 2 ·

Basic reporting

The manuscript proposes a message recovery attack on SBIM(Q), a multivariate polynomial trapdoor over the field of rational numbers Q, which was proposed by Markovski et al in 2014. SBIM(Q) encrypts a plaintext of n rational numbers into a ciphertext of 2n rational numbers, and its security is based on the complexity of 2n polynomial equations with 3n unknowns.
The authors of this manuscript demonstrate a successful cryptoanalysis on the SBIM(Q) trapdoor using a Grobner basis attack. They show that plaintext message recovery from the ciphertext can be achieved without knowing the secret key. The authors use a toy example from the original paper to demonstrate the cryptanalysis and find that some of its Grobner basis only consist of information variables. This information-only polynomial in the Grobner basis can be used to retrieve the secret. The authors point out that the cost of this vulnerability in SBIM(Q) is due to the linear bipermutations, which are exploited by the Grobner basis attack.The paper is well-written with sufficient information and background and good logic description.
The revisions meet my comments so I recommend accepting it now.

Experimental design

The author have revised the manuscript and it reaches to the acceptable level.

Validity of the findings

this is a revised version.